# Women's knowledge and its associated factors regarding optimum utilisation of antenatal care in rural Ghana: A cross-sectional study

**Agani Afaya**[1]*, **Thomas Bavo Azongo**[2], **Veronica Millicent Dzomeku**[3], **Richard Adongo Afaya**[3], **Solomon Mohammed Salia**[1], **Peter Adatara**[1], **Robert Kaba Alhassan**[4], **Abigail Kusi Amponsah**[3], **Confidence Alorse Atakro**[5], **David Adadem**[1], **Emmanuel Opoku Asiedu**[1], **Paul Amuna**[6], **Martin Amogre Ayanore**[6]

**1** Department of Nursing, School of Nursing and Midwifery, University of Health and Allied Sciences, Ho, Ghana, **2** Department of Public Health, School of Allied Health Sciences, University for Development Studies, Tamale, Ghana, **3** Department of Nursing, Kwame Nkrumah University of Science and Technology, Kumasi, Ghana, **4** Centre for Health Policy and Implementation Research. Institute of Health Research, University of Health and Allied Sciences, Ho, Ghana, **5** School of Nursing and Midwifery, Christian Service University College, Kumasi, Ghana, **6** School of Public Health, University of Health and Allied Sciences, Ho, Ghana

* aagani@uhas.edu.gh

## Abstract

### Introduction

Improving maternal health is a global public health challenge especially in sub-Saharan Africa. The optimum utilisation of antenatal care (ANC) by pregnant women is known to improve maternal health outcomes. Maternal morbidity and mortality rates in Ghana remain unacceptably high, particularly in rural settings where skilled delivery care often times is disproportionally low. This study assessed factors associated with optimum utilisation of antenatal care in rural Ghana.

### Methods

A cross-sectional design was applied to collect data among eligible participants between October 2018 and January 2019. A total of 322 women who gave birth and attended the postnatal clinic were recruited for the study. Consecutive sampling was employed in recruiting participants. The associations between the dependent variables (ANC service utilisation and knowledge of ANC) and independent variables (socio-demographic characteristics) were examined using ordinary least squares logistic regression at 95% confidence interval in STATA version 14.0.

### Results

Of the 322 participants, 69.0% reported utilising at least four or more times ANC services. Determinants of women attending ANC for four or more times was significantly associated with age [OR = 4.36 (95%CI: 2.16–8.80), p<0.001], educational level [OR = 10.18 (95%CI:

**Data Availability Statement:** All relevant data are within the manuscript and its Supporting Information files.

**Funding:** The author(s) received no specific funding for this work.

**Competing interests:** The authors have declared that no competing interests exist.

**Abbreviations:** ANC, Antenatal care; CHPS, Community-based Health Planning Services; EN, Enrolled Nurses; GMHS, Ghana Maternal Health Survey; MMR, Maternal Mortality Ratio; NHIS, National Health Insurance scheme; SBCC, Social and Behaviour Change Communication; WHO, World Health Organisation.

3.86–26.87), p<0.001], and insured with National Health Insurance Scheme [OR = 3.42 (95%CI: 1.72–6.82), p<0.001]. Not married [OR = 0.65 (0.39–1.09), p = 0.011] or divorced [OR = 0.33 (95%CI: 0.13–0.83), p = 0.019] was negatively associated with utilisation of four or more ANC services. The majority (79.0%) of the participants had a good level of knowledge regarding antenatal care.

## Conclusion

Although the majority of women in this study had good knowledge of ANC services, a significant number of them did not complete the recommended number of ANC visits for at least four times during a normal pregnancy. Awareness and further education to reproductive-age women on the significant role adequate ANC attendance plays in advancing health and well-being require further investments, particularly among rural women in Ghana.

## Introduction

Globally, about 303, 000 women and adolescent girls die as a result of pregnancy and childbirth-related complications [1] of which an alarming 99% of these maternal deaths occur in low-resource settings. Though globally, the number of women and girls dying due to pregnancy-related complications and childbirth decreased by nearly half (50%) from 1990 to 2013, the number of deaths within the West African region remains unacceptably high with Maternal Mortality Ratio (MMR) of 679 deaths per 100,000 live births in 2015 [2–4].

According to the 2017 Ghana Maternal Health Survey (GMHS), the MMR in Ghana is 310 per 100,000 live births, which remains unacceptably high. Data from the 2017 GMHS shows that nearly half of the women (50%) in rural settings of the Northern and Volta regions of Ghana deliver at home compared with the national average of 21% [5]. Delivering at home without the presence of a skilled birth attendant predisposes pregnant women to some life-threatening conditions. Facility-based delivery allows pregnant women to receive emergency care by skilled health professionals when complications arise which can help save the lives of both baby and mother [5]. Optimum antenatal care utilisation can pre-empt women the need for hospital delivery, thereby reducing possible pregnancy-related morbidities and mortalities.

Antenatal care (ANC) is the health care and education provided to pregnant women and adolescent girls by skilled health care professionals to ensure the best health conditions for the mother and the baby during pregnancy [1, 5]. Antenatal care utilisation is an important constituent of maternal health care, which reduces maternal and perinatal morbidity and mortality both directly through identification and management of pregnancy-related complications, and indirectly through the identification of pregnant women and girls most likely to develop complications during labour and delivery, thus ensuring early referral to an appropriate health facility for further care [1]. Globally, while most women now attend at least one ANC visit (86%), only 62% attend four, with lower rates reported in sub-Saharan Africa and South Asia [6]. An analytical review of the recent WHO Global Health Observatory data repository shows that ANC coverage, between 2000 and 2017, was indirectly correlated with MMR worldwide [7, 8]. The evidence indicates that countries with poor ANC coverage are more likely to have high MMR [9–11]. For example, ANC utilisation in Australia is 94% with MMR of 6 per 100, 000, Finland has 99% ANC utilisation with MMR 3 per 100, 000 and France has 99% ANC utilisation with MMR 8 per 100000 live births. In comparison with sub-Saharan Africa, Nigeria

has 49.1% ANC coverage with MMR of 917 per 100000, Cote d'Ivoire has 51.3% ANC coverage with MMR of 617 per 100,000, and Ghana has 89.3% ANC coverage with MMR 310 deaths per 100, 000 live births [7,8].

While ANC coverage remains high in Ghana, the coverage of at least four ANC visits remains lower at approximately 76% [12]. During pregnancy, the WHO recommends at least 4 ANC visits for antenatal care by a skilled health care professional for advice and monitoring of the health and well-being of both the mother and the developing foetus. A minimum of 4 visits constitutes receiving optimum ANC care. In this study, optimum ANC refers to a woman who made at least 4 visits for ANC during pregnancy. Geographically, despite the national rate of 76% for ANC, disparities exist particularly among rural and poor socio-economic groups in Ghana on health care utilisation. To advance maternal and reproductive health outcomes for women throughout their life cycle and contribute to reducing current MMR, there is a need for more context evidence across Ghana in relation to an important outcome measure such as pregnant women's knowledge and optimum ANC during pregnancy. This study, therefore, sought to assess women's knowledge and its associated factors that influence optimum utilisation of ANC services (4+ visits) in a context that presents opportunity for health systems learning and policy recommendations as Ghana steps up efforts to ensure universal access to reproductive health care and for the attainment of the Sustainable Development Goals (SDGs) targets on maternal and reproductive health.

## Conceptual framework

Deciding to utilise ANC services does not only depend on the individual's preference but also the factors that facilitate the accessibility of the health facility (services) or the factors that impede the utilisation of ANC services [13]. WHO outlined that a woman's ability to access antenatal care services is influenced by several factors that include distance or time taken to travel to a health facility, availability of ANC services, social and cultural factors that can serve as barriers to access, economic status, costs related with utilisation of services, and the ANC quality of services provided [14].

Several models have been developed over the last few decades to identify factors influencing healthcare service utilisation, and the most widely used is the Anderson Behavioral Health care model known as Behavioral Model of Health Services. This study was grounded based on the Andersons Behavioral Model and sought to identify factors that facilitate or impede the use of healthcare and to understand discrepancies in the utilisation of care [15]. This model was first developed to determine why families utilise health care, to identify equitable access to health services, and to support in formulating policies that will promote equitable access [16, 17]. According to the model, utilisation of health services is determined by three dynamics: (1) the predisposing factors, (2) enabling factors, and (3) need factor. Predisposing factors include individual characteristics and health beliefs; enabling factors are the logistic aspect of obtaining care and need factors are the most immediate cause of healthcare utilisation such as the presence of disease condition and self-perceived functional or health status [16].

The current study sought to assess factors that influence (facilitate or impede) women's use of ANC services; community-based factors that impede or motivate the use of ANC, facility-based factors, and adequacy of ANC visits. Despite the importance of these individual determinants affecting ANC utilisation among women, the current study also sought to assess women's knowledge of ANC services and how that influence ANC use as evident in literature [18] recommended by WHO for pregnant women. The study identified the following (as shown in Fig 1) factors that influence ANC utilisation (either positive or negative) in the study setting among women.

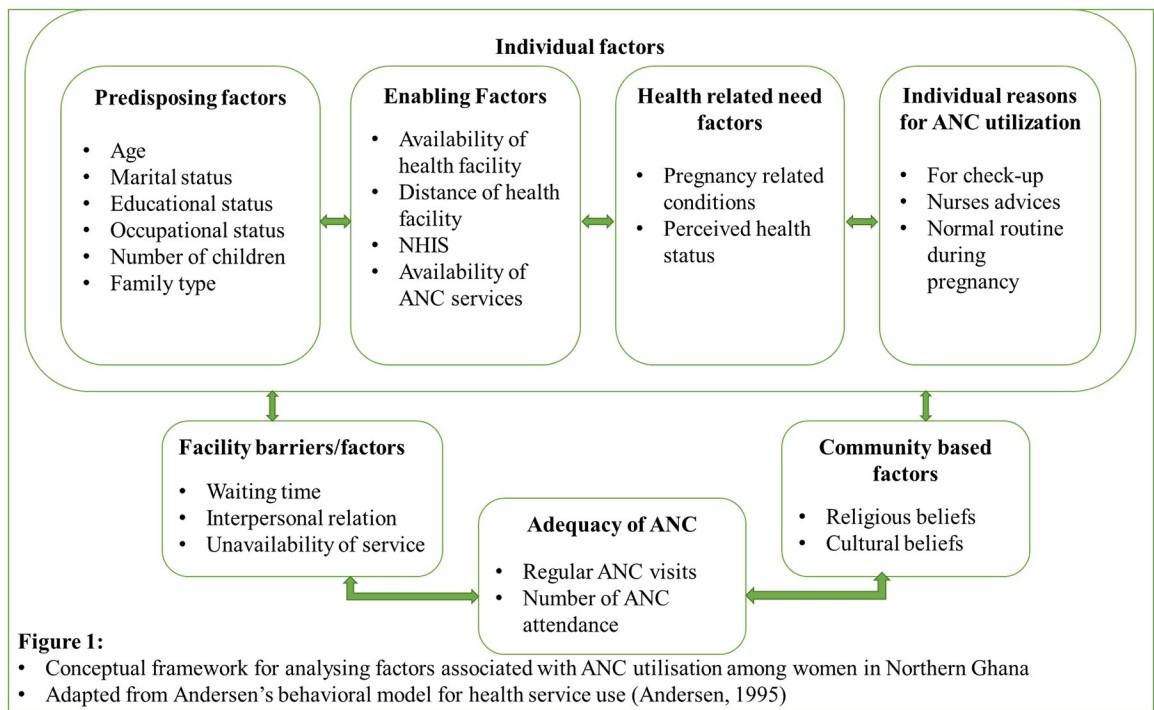

**Figure 1:**
- Conceptual framework for analysing factors associated with ANC utilisation among women in Northern Ghana
- Adapted from Andersen's behavioral model for health service use (Andersen, 1995)

**Fig 1. Conceptual framework for analysing factors associated with ANC utilisation among woman in Northern Ghana.** Adapted from Andersen's behavioral model for health service use (Anderson, 1995).

## Methods

### Study design

The study applied a cross-sectional analytical facility-based study design to recruit the study participants. Given the design type, quantitative data was collected among eligible study participants after applying the appropriate sampling methodology. This design was appropriate to enable the study team to elicit answers to key questions that underpin the study.

### Study setting and population

The study was conducted at four health facilities in Bamboi, a sub-district jurisdiction under the Bole district in the Savanah region of Ghana. The health facilities were categorised as (A, B, C & D). Facility A had two midwives, four enrolled nurses (EN) and a community health nurse (CHN), facility B had two CHNs, facility C had a midwife and one CHN and facility D had male practicing midwife. These health professionals provide maternal and child health services for mothers attending any of these facilities. The Bole district is one of the seven districts located in the newly created Savanah region of Ghana. The district is located at the extreme western part of the northern region of Ghana. The district is bordered to the north by Sawla / Tuna Kalba District, to the West by the Republic of Ivory Coast, to the east by West Gonja District and to the south by Wenchi and Kintampo Districts of Brong-Ahafo Region. The district health services are divided into four sub-districts namely Bole, Tinga, Jama, and Bamboi. Each sub-district has an operational area served by a health facility. Bamboi is one of the sub-districts which is to the South of the Bole district. The sub-district, Bamboi where this study was conducted has a relatively small semi-urban area while a large proportion is rural. The Bamboi sub-district has one health center and three Community-Based Health Planning and Services

(CHPS) zones serving over 26 communities within the sub-district. In Ghana, CHPS is the lowest level under the health level structures where health care is provided at the community level, closer to the people. The eligible study population were all reproductive-aged women in the Bambi sub-district who attended the study health facilities in Bamboi sub-district for post-natal care during the time of the study.

## Sampling method and sample size determination

Consecutive sampling was employed in recruiting participants. The average monthly attendance for the four facilities providing postnatal care services was estimated to be about 450. The total population for the four facilities for four months was 1800. The sample size for the study was determined by using the sample size calculation by Yamane [19]. The minimum required sample size was 310 and a 5% non-response rate was added to get a sample size of 340. Finally, 322 questionnaires were considered valid to be included for analysis.

## Outcome variables

During data analysis, the outcome variables of interest were knowledge and utilisation of ANC services. The utilisation of ANC services was classified as adequate versus inadequate. The study assessed the overall adequacy of ANC visits by reviewing the mothers' ANC cards to determine the frequency of visits during the immediate past pregnancy. Antenatal care was classified as adequate if the woman had at least four or more antenatal visits during normal pregnancy and less than four visits were defined as inadequate. The knowledge of mothers on ANC services was classified as good knowledge and bad knowledge.

## Explanatory variables

The explanatory variables were selected based on review of related literature and the modified version of the Anderson Healthcare Utilisation model. Socio-demographic characteristics; Participants' age was categorised into <20yrs, 21-30yrs and 31–46 years. Marital status was categirised into not married, and married. Educational level of participants was categorised as none, basic, secondary and tertiary. Occupation of participants was classified into farmer, trader, civil servant, and others. The religious status was categorised under Christianity and Islam. Mo, Gonja, Ewe, Akan, and Dagaaba were the ethnic groups of participants. The family type was the nuclear and extended family. The number of children was categorised into 1–2, 3–4 and 5+ children. Participants were either insured or not insured with the National Health Insurance Scheme (NHIS). Other variables of interest were distance to health facility (For CHPS, the distance was between a radius of 5 km (and for Health Centres between 8 and 16 km radius [20–22]. Women who trekked more than 5km and also 16km where considered far and those within the recommended limit were considered short), and means of transportation to the facility.

## Data collection procedure

Data collection lasted for four months from October 2018 to January 2019. A semi-structured and pretested questionnaire was used to collect data from eligible participants. Four trained research assistants (nurses) assisted to collect data in the four study health facilities. The research assistants were trained for 2 days and were supervised for a week by the principal investigator during data collection. The four facilities were visited every week during the post-natal days for questionnaire administration. Eligible participants were approached after receiving postnatal care and provided with further education on the study's aim and objectives.

Eligible participants who were willing to participate were offered consent forms to endorse. Questionnaires were interviewer-administered by the study team. Completeness of questionnaires was checked daily after completing each questionnaire before participants were asked to leave. Where minor omissions were detected, immediate corrective measures were taken with participants to ensure validity and to prevent missing information.

## Validity and reliability of the study instrument

The study instrument was developed after reviewing relevant related literature and also adopting Anderson health care behavioral model. Preceding data collection, the validity and reliability of the study instrument were assessed through various stages. To assess the content validity, a group of researchers and clinicians were made to review and refine the items in the questionnaire. This group comprised of two senior community midwives and two researchers in the field of maternal and child health. Further, the Pearson correlation coefficient was used to determine the test-retest reliability of the study with a sample of 20 participants. The instrument was administered to the same participants' after one week. The test-retest correlation coefficient for the instrument was 0.73 (p < 0.001), which is indicative of acceptable stability over time.

## Ethical consideration

Ethics approval for the study was obtained from the University of Health and Allied Sciences Research Ethical Committee (UHAS-REC A. 10[27] 17–18). Permission to conduct the study was sought from the various facilities before the commencement of data collection. Both verbal and written consent was obtained from participants after agreeing to participate in the study. Participants were assured of anonymity, confidentially and privacy. Participants were made aware of the right to withdraw from the study at any particular point in time if they wished to do so.

## Data analysis

Data were entered into SPSS software, cleaned and validated to ensure completeness and quality before analysis. The cleaned data were exported to STATA version 14.0 for analysis. Descriptive statistics were applied to represent frequencies and percentages. The overall level of knowledge on antenatal care was measured by scoring responses that measured participants' knowledge (Box 1) using the following descriptions; a score of 1 was assigned if the participant had knowledge and 0 if they had no knowledge. Total score and mean scores were computed and a score below the mean was considered poor knowledge whiles a mean score or above was considered to be a good knowledge. Ordinary least square logistic regression analysis, computing odds ratio, was used to determine the association between dependent variables (ANC service utilisation and knowledge of ANC) and socio-demographic characteristics. Statistical significance was determined at 95% confidence level.

## Results

### Participants' socio-demographic characteristics

Slightly more than half, 177 (55.0%) of participants were aged 21–30 years while about half, 167 (51.9%) were not married. A quarter of the women, 111 (34.5%) had basic and secondary education, and more than half, 188 (58.4%) engaged in trading. About 144 (44.7%) belonged to the Dagaaba ethnic group. The majority, 288 (89.4%) had the nuclear family type and over

> **Box 1. Key items used to assess women's knowledge of ANC services**
>
> **Questions asked on ANC knowledge**
>
> 1. ANC can prevent complications in pregnancy?
>
> 2. Pregnant women may have problems without ANC?
>
> 3. Regular ANC medications can promote optimal growth of unborn child?
>
> 4. Health facility delivery is safer and better than home delivery?
>
> 5. Do you know the recommended place for delivery?
>
> 6. Do you know the recommended frequency and timing of ANC visits?
>
> 7. ANC is recommended regardless of complications?
>
> 8. Do you know items to prepare for before delivery?
>
> 9. Do have knowledge of family planning?
>
> 10. Do have knowledge of malaria prevention?

two-thirds had 1–2 children. The majority, 277 (86.1%) had to walk to access healthcare. Most, 210 (65.2%) made their own decision regarding their healthcare needs (Table 1).

## Factors associated with the utilisation of antenatal care services

Logistic regression analysis showed that women who were aged 21–30 years were 4 times more likely to utilise ANC services four or more times compared to those aged 20 years and below [OR = 4.36 (95%CI: 2.16–8.80), p<0.001] and those who were not married were 70% less likely to utilise ANC services at least four times compared to those who were married [OR = 0.30 (95%CI: 0.18–0.50), p<0.001]. Women who had tertiary education were 10 times more likely to utilise ANC services four or more times than those without education [OR = 10.18 (95%CI: 3.86–26.87) p<0.001]. Participants who were insured with the NHIS were 3 times more likely to utilise the ANC compared to those who were not-insured [OR = 3.42 (95%CI: 1.72–6.82), p<0.001]. Regarding distance to health facilities, women who reported that they stayed far from health facilities were 80% less likely to utilise ANC compared to women who stayed closer to health facilities [OR = 0.20 (95%CI: 0.12–0.34), p<0.001] (Table 2).

After adjusting for other explanatory variables women who were not married were 84% less likely to utilise ANC services four or more times [AOR = 0.16 (95%CI: 0.01–0.75), p = 0.03]. Women who had tertiary education were 10 times more likely to utilise ANC services four or more times than those without education [AOR = 5.26 (95%CI: 2.15–17.28) p<0.001]. Women who stayed far from the health facilities were 65% less likely to attend ANC for at least four times [AOR = 0.35 (95%CI: 0.02–0.13), P = 0.001] (Table 2).

## Participants knowledge of antenatal care services

Fig 2 shows that 79.2% of the participants had good level of knowledge regarding of antenatal care while 20.8% had poor knowledge regarding antenatal care.

**Table 1. Participants' socio-demographic characteristics (N = 322).**

| Variable | Response | Frequency (N) | Percentage (%) |
|---|---|---|---|
| **Age** | ≤ 20 | 44 | 13.6 |
| | 21–30 | 177 | 55.0 |
| | 31–40 | 101 | 31.4 |
| **Marital Status** | Married | 133 | 41.3 |
| | Not married | 189 | 58.7 |
| **Educational status** | None | 33 | 10.2 |
| | Basic | 111 | 34.5 |
| | Secondary | 111 | 34.5 |
| | Tertiary | 67 | 6.8 |
| **Employment** | Employed | 122 | 37.9 |
| | Unemployed | 189 | 58.7 |
| | Student | 11 | 3.4 |
| **Occupation** | Farmer | 56 | 17.4 |
| | Trader | 188 | 58.4 |
| | Civil servant | 56 | 17.4 |
| | Others | 22 | 6.8 |
| **Religion** | Christianity | 255 | 79.2 |
| | Islam | 67 | 20.8 |
| **Ethnicity** | Mo | 100 | 31.1 |
| | Gonja | 33 | 10.3 |
| | Ewe | 11 | 3.4 |
| | Akan | 34 | 10.5 |
| | Dagaaba | 144 | 44.7 |
| **Family type** | Nuclear | 288 | 89.4 |
| | Extended | 34 | 10.6 |
| **Number of children** | 1–2 children | 245 | 76.1 |
| | 3–4 children | 55 | 17.1 |
| | 5+ children | 22 | 6.8 |
| **Number of ANC visits** | <4 | 100 | 31.0 |
| | 4+ | 222 | 69.0 |
| **Health insurance** | Insured | 245 | 76.1 |
| | Not insured | 77 | 23.9 |
| **Distance to health facility** | Short | 220 | 68.3 |
| | Far | 102 | 31.7 |
| **Transportation to ANC** | Motorcycle | 23 | 7.1 |
| | Foot | 277 | 86.1 |
| | Others | 22 | 6.8 |
| **Person who provides for healthcare** | Self | 90 | 27.9 |
| | Partner | 166 | 51.6 |
| | Parents | 66 | 20.5 |
| **Person who makes healthcare decision** | Self | 210 | 65.2 |
| | Partner | 101 | 31.4 |
| | Parents | 11 | 3.4 |

**Table 2. Logistic regression results showing optimum utilisation of ANC services.**

| Variable | Response | Usage of ANC Services | | COR [95% CI] | p-value | AOR [95% CI] | p-value |
|---|---|---|---|---|---|---|---|
| | | <4 = 100 n(%) | 4+ = 222 n(%) | | | | |
| **Age** | ≤20 | 22 (22.0) | 22 (9.9) | Ref | | Ref | |
| | 21–30 | 33 (33.0) | 144 (64.9) | 4.36[2.16–8.80] | <0.001 | 2.01[0.32–5.70] | 0.043 |
| | 31–46 | 45 (45.0) | 56 (25.2) | 1.24[0.61–2.53] | 0.546 | 0.24[2.16–8.80] | 0.067 |
| **Marital Status** | Married | 33 (33.0) | 100 (45.0) | Ref | | | |
| | Not married | 67 (67.0) | 122 (55.0) | 0.30 [0.18–0.50] | <0.001 | 0.16 [0.01–0.75] | 0.030 |
| **Educational status** | None | 22 (22.0) | 11 (5.0) | Ref | | Ref | |
| | Basic | 22 (22.0) | 89 (40.1) | 8.09 [3.42–19.14] | <0.001 | 3.19 [2.35–16.75] | 0.056 |
| | Secondary | 45 (45.0) | 66 (29.7) | 2.93 [1.29–6.64] | 0.010 | 1.29 [1.29–6.64] | 0.045 |
| | Tertiary | 11 (11.0) | 56 (25.2) | 10.18[3.86–26.87] | <0.001 | 5.26[2.15–17.28] | <0.001 |
| **Occupation** | Farmer | 33 (33.0) | 23 (10.4) | Ref | | | |
| | Trader | 45 (45.0) | 143 (64.4) | 4.55[2.40–8.55] | <0.001 | | |
| | Civil servant | 11 (11.0) | 45 (20.3) | 5.87[2.51–13.69] | <0.001 | | |
| | Others | 11 (11.0) | 11 (4.9) | 1.43[0.53–3.86] | 0.475 | | |
| **Religion** | Christianity | 44 (44.0) | 211 (95.1) | Ref | | Ref | |
| | Islam | 56 (56.0) | 11 (4.9) | 0.04[0.02–0.08] | <0.001 | 0.017[0.01–0.29] | 0.088 |
| **Family type** | Nuclear | 88 (88.0) | 200 (90.1) | Ref | | | |
| | Extended | 12 (12.0) | 22 (9.9) | 0.81[0.38–1.70] | 0.573 | | |
| **Health insurance** | Not Insured | 11 (11.0) | 66 (29.7) | Ref | | | |
| | Insured | 89 (89.0) | 156 (70.3) | 3.42[1.72–6.82] | <0.001 | | |
| **Distance to health facility** | Short | 44 (44.0) | 176 (79.3) | Ref | | Ref | |
| | Far | 56 (56.0) | 46 (20.7) | 0.20[0.12–0.34] | <0.001 | 0.35[0.02–0.13] | 0.001 |
| **Provider for healthcare** | Self | 34 (34.0) | 56 (25.2) | Ref | | | |
| | Partner | 33 (33.0) | 133 (59.9) | 2.44 [1.38–4.33] | 0.002 | | |
| | Parents | 33 (33.0) | 33 (14.9) | 0.60 [0.32–1.16] | 0.129 | | |

**COR** = Crude Odds ratio, **AOR** = Adjusted Odds ratio, adjusted for age, marital status, educational level, religion, and distance to health facility.

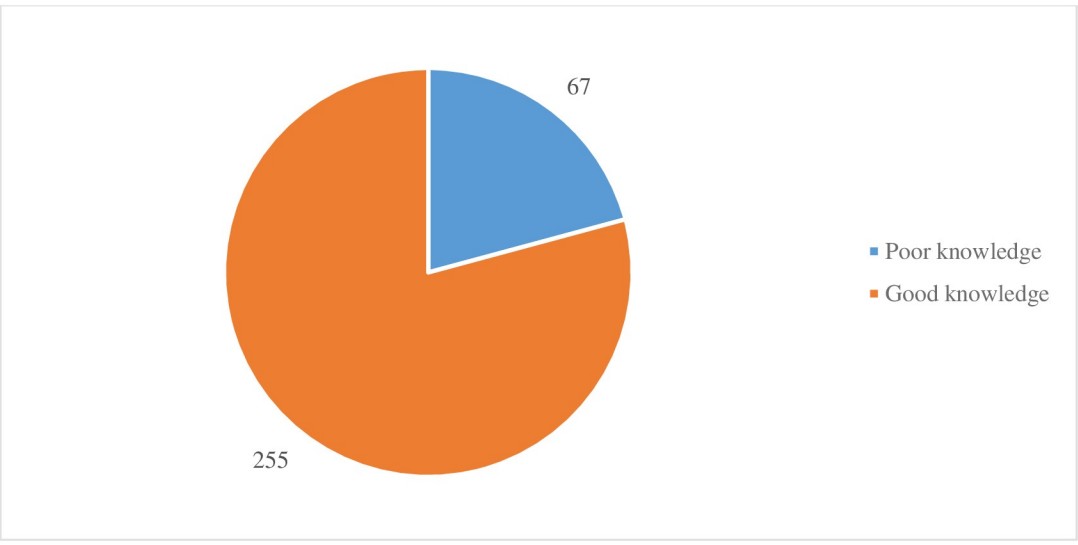

**Fig 2. Overall level of knowledge regarding ANC.**

## Association between Socio-demographics characteristics of participants and knowledge of antenatal care

Not being married and divorced women were 75% and 91% respectively less likely to have good knowledge regarding ANC [OR = 0.25, (95%CI: 0.13–0.51), p<0.001; OR = 0.09, (95% CI: 0.03–0.26), p<0.001]. Women who attained basic school education were 4.47 times more likely to have good knowledge of ANC compared to those with no education [OR = 4.47, (95%CI: 1.75–11.39), p = 0.002]. Respondents who belonged to the Islamic religion were 85% less likely to have good knowledge on ANC compared to respondents who were Christians [OR = 0.15, (95%CI: 0.08–0.27), p<0.001]. Regarding family status and its association with women use of ANC services, women who lived in an extended family system were 99% less likely to have good knowledge on ANC compared to those who lived in a nuclear family environment [OR = 0.01, (95%CI: 0.00–0.03), p< 0.001] (see Table 3).

Multi-variable logistic regression analysis revealed that respondents aged >30 years were 98% less likely to have adequate knowledge on antenatal care compared to those aged 20 years and below [AOR = 0.02, (95%CI: 0.00–0.45), p = 0.013]. Not married women were 99% less likely to have adequate knowledge on antenatal care compared to married women [AOR = 0.01, (95%CI: 0.00–0.28), p<0.001]. Women who had 5+ children were 98% less

**Table 3. Logistic regression results of factors associated with women's knowledge of ANC services.**

| Variable | | Knowledge of ANC Services | | COR [95% CI] | p-value | AOR[95% CI] | p-value |
|---|---|---|---|---|---|---|---|
| | | Poor n = 67 (%) | Good n = 255 (%) | | | | |
| **Age** | ≤ 20 | 0 (0.00) | 44 (17.2) | Ref | | Ref | |
| | 21–30 | 33 (49.3) | 144 (56.5) | 0.05 [0.00–0.81] | 0.035 | 0.10 [0.01–1.76] | 0.115 |
| | >30 | 34 (50.7) | 67 (26.3) | 0.02 [0.00–0.36] | 0.008 | 0.02 [0.00–0.45] | 0.013 |
| **Marital Status** | Married | 11 (16.4) | 122 (47.8) | Ref | | Ref | |
| | Not married | 56 (83.6) | 1323(52.2) | 0.25[0.13–0.51] | <0.001 | 0.01 [0.00–0.28] | <0.001 |
| **Educational status** | None | 11 (16.4) | 22 (8.6) | Ref | | | |
| | Basic | 11 (16.4) | 100 (39.2) | 4.47[1.75–11.39] | 0.002 | | |
| | Secondary | 34 (50.8) | 77 (30.2) | 1.15[0.51–2.60] | 0.740 | | |
| | Tertiary | 11 (16.4) | 56 (22.0) | 2.51[0.96–6.50] | 0.058 | | |
| **Occupation** | Farmer | 11 (16.4) | 45 (17.6) | Ref | | | |
| | Trader | 45 (67.2) | 43 (56.1) | 0.80[0.38–1.65] | 0.541 | | |
| | Civil servant | 11 (16.4) | 45 (17.7) | 0.10 [0.40–2.50] | 0.001 | | |
| | Others | 0 (0.00) | 22 (8.6) | 11.37[0.64–21.83] | 0.098 | | |
| **Religion** | Christianity | 33 (49.3) | 222 (87.1) | Ref | | Ref | |
| | Islam | 34 (50.7) | 33 (12.9) | 0.15[0.08–0.27] | <0.001 | 0.23 [0.08–0.71] | 0.010 |
| **Ethnicity** | Mo | 0 (0.00) | 100 (39.2) | Ref | | | |
| | Gonja | 22 (32.8) | 11 (4.3) | 0.01[0.00–0.04] | <0.001 | | |
| | Ewe | 0 (0.00) | 11 (4.3) | 0.11[0.00–6.04] | 0.280 | | |
| | Akan | 12 (17.9) | 22 (8.7) | 0.01[0.00–0.16] | 0.001 | | |
| | Dagaaba | 33 (49.3) | 111 (43.5) | 0.02[0.00–0.27] | 0.004 | | |
| **Family type** | Nuclear | 33 (49.2) | 255 (100) | Ref | | Ref | |
| | Extended | 34 (50.7) | 0 (0.00) | 0.01 [0.00–0.03] | <0.001 | 0.01 [0.00–0.02] | <0.001 |
| **Number of children** | ≤ 2 children | 45 (67.1) | 200 (78.4) | Ref | | Ref | |
| | 3–4 children | 11 (16.4) | 44 (17.3) | 0.89 [0.42–1.81] | 0.725 | 0.02 [0.00–2.82] | 0.116 |
| | 5+ children | 11 (16.4) | 11 (4.3) | 0.23 [0.09–0.55] | 0.001 | 0.02 [0.00–0.43] | 0.012 |

**COR** = Crude Odds ratio, **AOR** = Adjusted Odds ratio, Adjusted for age, marital status, religion, family type, number of children.

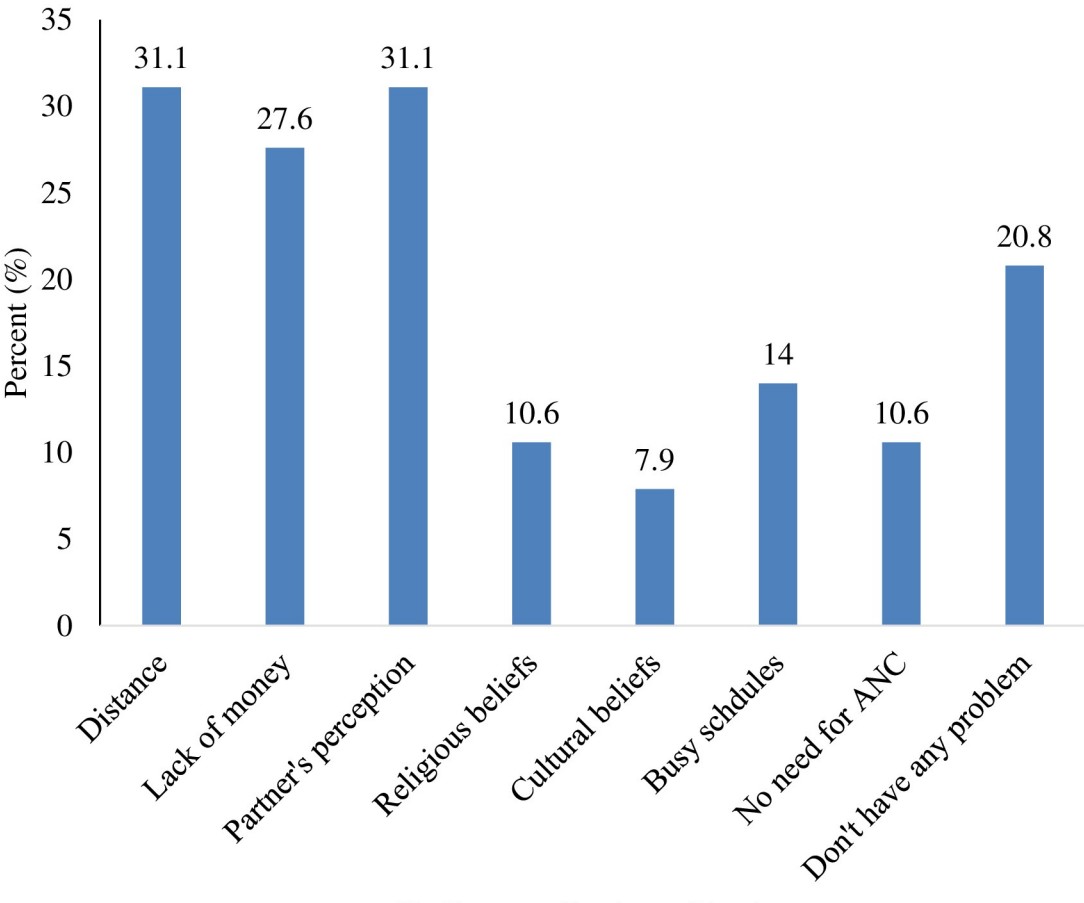

**Fig 3. Challenges affecting utilisation of ANC.**

likely to have adequate knowledge on antenatal care compared to those with 2 children or less [AOR = 0.02, (95%CI: 0.00–0.43), p = 0.012].

## Community and individual challenges affecting antenatal utilisation

Long-distance to the health facility (31.1%) and partner's perception of the importance of ANC were major barriers to the effective utilisation of ANC services by the women. Also, about 27.6% indicated not having enough money to attend antenatal clinic and cultural beliefs (7.9%) as some of the challenges. This is shown in (Fig 3).

## Facility related barriers to ANC attendance

Fig 4 showed that most, 38.2% of the participants were faced with long waiting times in health facilities where they assessed care, followed by 28% who were scolded by staff and 24.5% who had inconvenient service hours.

## Reasons for ANC attendance

Fig 5 also showed that majority, 76.1% of the participants went for antennal services for check-ups, while in 24.2% of them, it was because of normal routine check-ups during pregnancy.

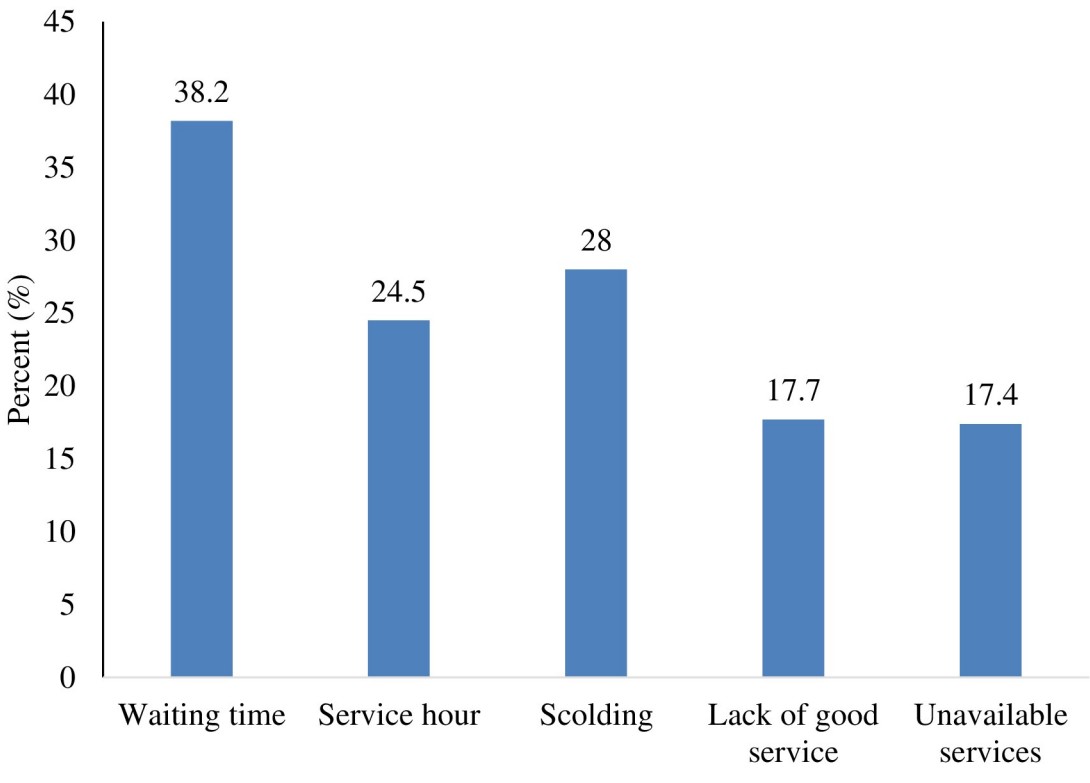

**Fig 4. Facility related barriers to ANC utilisation.**

Also, about one-tenth, 10.6% went for antenatal care services as a result of health workers' advice.

## Discussion

This study applied an integrated conceptual framework to identified individual and facility based factors that influence optimum utilisation of ANC services among mothers in rural areas. Adequacy of ANC services was also assessed. And for a pregnant woman to attain optimum utilisation or adequacy of ANC services, the World Health Organization recommends four or more ANC visits for women who do not experience any complications throughout their pregnancies [23]. The current study finding revealed that 69.0% of the women attended ANC at least four or more times. This study corroborates results in Ethiopia [24] but higher than some earlier studies in Rwanda [3], Ethiopia [25], and Kenya [26]. Though this study result is high as compared to other countries, it is low when compared to findings in previous studies in Ghana [27] Nepal [28] and Nigeria [9]. The divergence could be due to socio-cultural, economic, and awareness differences among the study populations as purported by other studies [25, 29]. The present finding of 69% is below the national coverage of at least four ANC visits of 75.9% [12]. This difference may be due to the lack of awareness of optimal ANC visits. Also, the differences could have been caused by other potential inhibiting factors such as distance to health facilities and socio-economic factors. Therefore, there is the need to intensify interventions implemented by the Ghana Health Service over the years such as Social and Behaviour Change Communication (SBCC) targeting reproductive-age women on

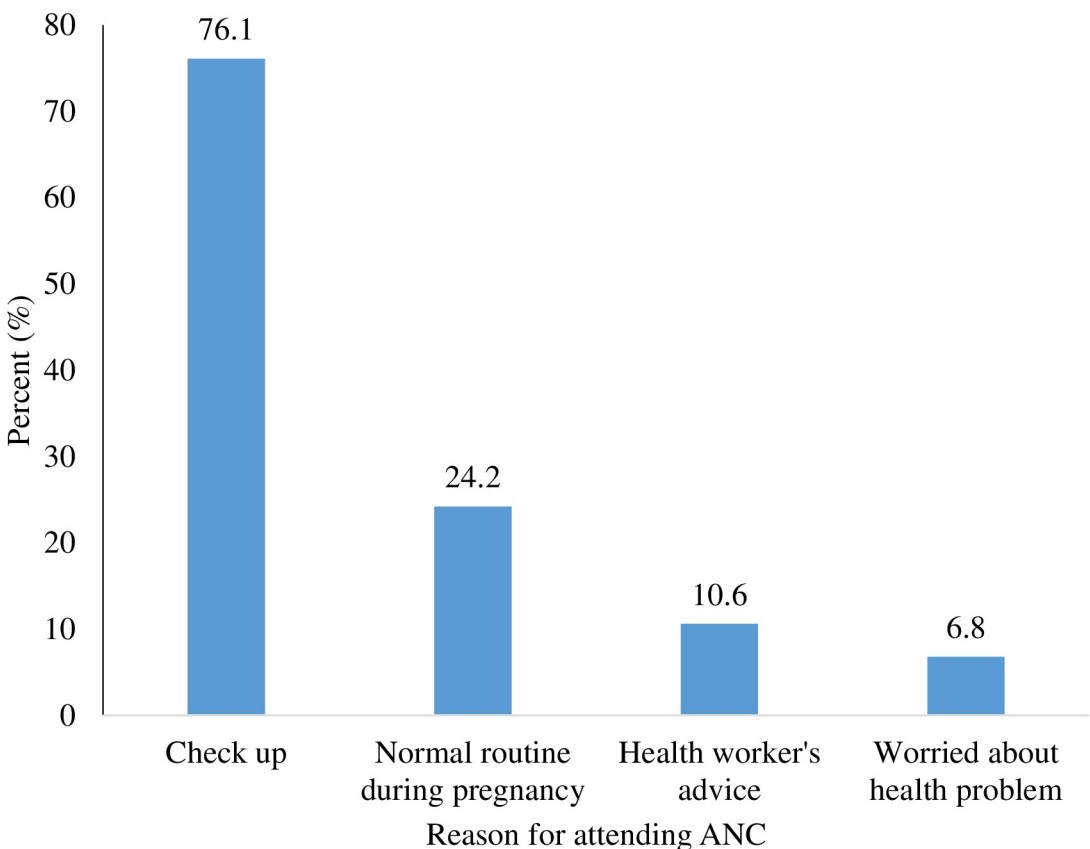

**Fig 5. Reason for attending ANC.**

optimum ANC utilisation and the importance of early initiation and regular ANC visits during pregnancy. In addition, current national policy initiatives to extend the Community-based Health Planning Services (CHPS) concept to cover the entire country in order to meet the reproductive health needs of many women, particularly in rural areas of Ghana, remain a laudable intervention in Ghana [12]. CHPS offer opportunity to bridge distance barriers and provide reproductive-age women better chances of obtaining relevant and useful information on ANC and its importance throughout pregnancy as well as serve as an effective Primary Health Care (PHC) strategy to improve access to maternal health and capacity building for improved client-service provider relationship.

Participants' educational level was found to be a strong predictor of optimum utilisation ANC services for four or more times. Participants who had basic, secondary, and tertiary education were more likely to use ANC services compared to those who had no education. This finding is not counter-intuitive because previous studies in developing countries have reported that the educational level of women increases the odds of utilising ANC services for four or more times [9, 23, 30–33]. Probably the educated women are more inclined to issues of their health (identifying danger signs) thereby placing more priority on their health and that of the unborn child. The major cause of health inequality in ANC coverage has been strongly associated with women's poor/lack of education [34]. This highlights the need for a multisectoral approach, with the health sector working jointly with other sectors such as the educational sector to promote female education to the highest level.

Participants who were not married or divorced were less likely to attend ANC for at least four or more times relative to their counterparts who were married. In a Ghanaian community, women who are not married are expected to remain virtuous until marriage [35] in order to prevent public ridicule, so women who get pregnant without a husband could avoid ANC visits [27] to prevent public ridicule. On the other hand, women who are not married and those with low socio-economic status are likely to lack the financial means (for travel cost and medicines) and may be worse affected when it comes to meeting the number of recommended ANC care services during pregnancy. The free fee delivery care policy introduced by the Government of Ghana through the NHIS continues to play a significant role in enhancing lower socio-economic groups' access and use of maternity care services, including ANC care. However, bottlenecks such as poor re-imbursements issues under the NHIS impact negatively on pro-poor women's access to ANC care and other general health care needs. Health system improvements that enhance the efficiency of the NHIS and broaden the benefits package for women reproductive health care services can go a long way to improve long term health gains for maternal health outcomes in Ghana.

Women who were insured with the NHIS were three times more likely to utilise ANC services for four or more times compared to those who were not insured. This finding is consistent with other studies in Ghana [27, 36–38]. While the number of visits a pregnant woman makes for ANC is key to attaining optimum ANC, the quality of ANC services is also relevant. Ayanore and colleagues in a study on focused ANC among women found that ANC quality predict the adequacy of ANC services relative to the frequency of visits made by a woman [39]. Ghana's implementation of a free delivery policy in July 2008 for pregnant women has played a major role in improving maternal and child health outcomes in Ghana. The NHIS which allows pregnant women access to benefit from this free fee policy has also assisted to improve health care access and utilisation for all age groups. This study findings show that being insured under the NHIS can enhance women's attainment of optimum ANC. Thus, the implementation of the free maternal health care policy under the NHIS instituted by the government of Ghana is key to achieving increased utilisation of ANC services for at least four times during every normal pregnancy [40].

Despite the potential benefit of the NHIS, registration challenges for enrolments often resulting in long queues making it difficult for pregnant women to register on time is documented [27, 41]. There is evidence in Ghana that demonstrates that poor NHIS client registration is mostly limited to the district capital because of the biometric system used; where people including pregnant women will have to trek long distances to register [27, 41]. The NHIS system has over the last few years introduced mobile phone technology to improve client registrations and address current challenges in enrolments under the NHIS. However, non-educated, pro-poor, and rural women are likely not to benefit from high technological applications on NHIS registrations. NHIS reforms must prioritize how to improve client's engagements for the rural poor, and enhance vulnerable groups' chances to obtain NHIS. New policy reforms that will enhance access for NHIS enrolments in distant and rural areas are recommended in addition to new technological strategies being currently implemented by the NHIS.

Women who trek far distances to the health facility for ANC services were 80% less likely to utilise ANC services for four or more times compared to those with trek shorter distances. The current study finding corroborates studies in Uganda and Pakistan where optimum ANC attendance was attributed to long distance to the health facility [42 43]. Establishment of health facilities within catchment areas in rural communities, employing more qualified health personnel to deliver healthcare to the women in the communities, and access to good roads among others by the government of Ghana is significant in ensuring increased ANC utilisation.

Women's knowledge of ANC is crucial in the utilisation of ANC services during pregnancy. This study revealed that 79.2% of the women had good knowledge of ANC services. The determinants of women having knowledge of ANC services included; woman's age, woman's educational status, husband's educational status, religion, ethnicity, and the number of children. Our study finding is consistent with a study conducted in the southwest of Nigeria [44] but disagrees with a study conducted in Mozambique where the researchers found women's knowledge of ANC services not to have significant importance in their utilisation of ANC services [45]. It is established that women with good knowledge of ANC services have a better understanding and acceptance of the services provided during ANC [46]. Women's chances to have a good knowledge of ANC services is thus a catalyst for improving ANC seeking. National-level policies that ensure focused ANC services, become an integral part of ANC and other maternal care services are fundamental to supporting improve knowledge, address myths and remove other health care seeking challenges women encounter when using ANC care services.

On the aspect of facility-based factors that hinders ANC utilisation, the study found that waiting for a long time in the facilities to receive care, was the major implicating factor for facility-based barriers to ANC utilisation. This finding is congruent with a study conducted in Mozambique where the women identified long waiting times as a barrier to ANC utilisation [45]. The long waiting time identified by participants as a barrier to ANC service utilisation could be due to the shortage of qualified healthcare professionals needed to provide care to over 26 communities in the study setting. Pregnant women might find it so distressing waiting for long hours without receiving care and forgoing certain duties which might have financial implications on them and this may demotivate women for subsequent ANC service utilisation. Gong et al., [45] further support our assertion by illustrating that long waiting times have direct and indirect cost implications for clients seeking care and that attending to clients on schedule mitigates cost and also improves women accessing ANC services.

## Limitations of the study

First, recall bias could have limited the validity of the data collected as some participants might have forgotten about past events involving ANC services. To minimize recall bias, the study assessed the immediate past ANC records and we crosschecked self-reported information against the ANC attendance book for the validity of the information. Secondly, the study design was cross-sectional, hence results should be interpreted with caution since this study did not provide an opportunity to assess cause-effect relationship. Also, several individuals, households, and health system factors influence optimum ANC services, which may have been missed and not included in this study. This study was facility-based, hence the opportunity to miss out on eligible participants at the population level is acknowledged. Notwithstanding, this study provides useful information on women's knowledge and factors that influence optimum antenatal care utilisation in rural settings in Ghana.

## Conclusion

Although the majority of the women had good knowledge of ANC services, a significant number did not complete the recommended number of ANC visits for at least four times during a normal pregnancy. The creation of awareness of adequate ANC attendance in rural settings should be emphasized by frontline health workers especially nurses and midwives working in these communities. Not insured with NHIS lowered the chances for women to have received optimum ANC services. Health system reforms that address NHIS access challenges are essential to support women attain optimum ANC and supportive focused ANC in Ghana.

Further research is recommended to determine sociocultural, traditional, religious and other related practices at the community level on the utilisation of antenatal care services.

## Supporting information

**S1 Data.**
(XLSX)

## Acknowledgments

Our profound gratitude goes to the health facilities for their support and co-operation during data collection. We are also grateful to the women who took time off their busy schedules to participate in the study.

## Author Contributions

**Conceptualization:** Agani Afaya, Thomas Bavo Azongo, Veronica Millicent Dzomeku, Richard Adongo Afaya, Solomon Mohammed Salia, Abigail Kusi Amponsah, David Adadem, Emmanuel Opoku Asiedu, Paul Amuna.

**Data curation:** Agani Afaya, David Adadem, Emmanuel Opoku Asiedu.

**Formal analysis:** Agani Afaya, Richard Adongo Afaya, Solomon Mohammed Salia, Peter Adatara, Robert Kaba Alhassan, Confidence Alorse Atakro, Paul Amuna, Martin Amogre Ayanore.

**Funding acquisition:** Agani Afaya.

**Investigation:** Agani Afaya, Thomas Bavo Azongo, Veronica Millicent Dzomeku, Richard Adongo Afaya, Peter Adatara, Robert Kaba Alhassan, Paul Amuna, Martin Amogre Ayanore.

**Methodology:** Agani Afaya, Thomas Bavo Azongo, Richard Adongo Afaya, Solomon Mohammed Salia, Peter Adatara, Robert Kaba Alhassan, Abigail Kusi Amponsah, Confidence Alorse Atakro, David Adadem, Emmanuel Opoku Asiedu, Paul Amuna, Martin Amogre Ayanore.

**Project administration:** Agani Afaya, Thomas Bavo Azongo, Veronica Millicent Dzomeku, Richard Adongo Afaya, Solomon Mohammed Salia, Peter Adatara, Robert Kaba Alhassan, Confidence Alorse Atakro, Paul Amuna, Martin Amogre Ayanore.

**Resources:** Agani Afaya, Richard Adongo Afaya.

**Software:** Agani Afaya, Richard Adongo Afaya.

**Supervision:** Agani Afaya, Thomas Bavo Azongo, Veronica Millicent Dzomeku, Richard Adongo Afaya, Peter Adatara, Robert Kaba Alhassan, Abigail Kusi Amponsah, Paul Amuna, Martin Amogre Ayanore.

**Validation:** Agani Afaya, Thomas Bavo Azongo, Veronica Millicent Dzomeku, Richard Adongo Afaya, Solomon Mohammed Salia, Peter Adatara, Robert Kaba Alhassan, Abigail Kusi Amponsah, Confidence Alorse Atakro, Paul Amuna, Martin Amogre Ayanore.

**Visualization:** Agani Afaya, Thomas Bavo Azongo, Veronica Millicent Dzomeku, Richard Adongo Afaya, Solomon Mohammed Salia, Peter Adatara, Abigail Kusi Amponsah, Confidence Alorse Atakro, Martin Amogre Ayanore.

**Writing – original draft:** Agani Afaya, Richard Adongo Afaya.

**Writing – review & editing:** Agani Afaya, Thomas Bavo Azongo, Veronica Millicent Dzomeku, Richard Adongo Afaya, Solomon Mohammed Salia, Peter Adatara, Robert Kaba Alhassan, Abigail Kusi Amponsah, Confidence Alorse Atakro, David Adadem, Emmanuel Opoku Asiedu, Paul Amuna, Martin Amogre Ayanore.

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
