## [Decision Letter · Decision Letter 0]

27 Mar 2020

PONE-D-20-05929

Women’s knowledge of and factors associated with the utilisation of antenatal care in rural Ghana: analysis of a community-based cross-sectional study

PLOS ONE

Dear Mr. Agani Afaya,

Thank you for submitting your manuscript to PLOS ONE. After careful consideration, we feel that it has merit but does not fully meet PLOS ONE’s publication criteria as it currently stands. Therefore, we invite you to submit a revised version of the manuscript that addresses the points raised during the review process.

We would appreciate receiving your revised manuscript by 30th May 2020. To enhance the reproducibility of your results, we recommend that if applicable you deposit your laboratory protocols in protocols.io, where a protocol can be assigned its own identifier (DOI) such that it can be cited independently in the future. For instructions see: http://journals.plos.org/plosone/s/submission-guidelines#loc-laboratory-protocols

We look forward to receiving your revised manuscript.

Kind regards,

Kwasi Torpey, MD PhD MPH

Academic Editor

PLOS ONE

Journal Requirements:

Reviewers' comments:

Reviewer's Responses to Questions

**Comments to the Author**

1. Is the manuscript technically sound, and do the data support the conclusions?

Reviewer #1: Yes

Reviewer #2: Partly

Reviewer #3: Partly

2. Has the statistical analysis been performed appropriately and rigorously? 

Reviewer #1: No

Reviewer #2: Yes

Reviewer #3: Yes

3. Have the authors made all data underlying the findings in their manuscript fully available?

Reviewer #1: Yes

Reviewer #2: No

Reviewer #3: Yes

4. Is the manuscript presented in an intelligible fashion and written in standard English?

Reviewer #1: Yes

Reviewer #2: Yes

Reviewer #3: Yes

5. Review Comments to the Author

Reviewer #1: This is an important study with a very clear focus.

Kindly find my comments below.

1. Line numbering would have made it easy to referencing.

2. The statement “The sub-district has semi-urban area but predominantly rural” under study design section should be put under study settings.

3. The variable “Number of Children” can’t be dichotomy when the categories are more than two. “The number of children was dichotomised into 1-2, 3-4 and 5+ children”

4. How was the distance to health facility measured? Distance from where to health facility? What distance is categorized as “far” and what is “near”?

5. “Those who sought ANC services less than four times were classified as having poor ANC service utilisation (inadequate) and those who had ANC visits of 4 or more were considered as having good (adequate) ANC service utilisation.” This information has been captured under the section on outcome variables already. Please delete it.

6. Revise this statement: “Bivariate logistic regression analysis, computing odds ratio, was used to determine the strength of the association…” Because Logistic regression will report odds ratio. Bivariate logistic regression analysis reports more than strength of association. What about direction of the association?

7. The statement “ANC service utilisation was coded as 0 for poor utilisation (<4 ANC visits) and 1 for good untilisation (4+ ANC visits)” has been repeated under the outcome variables and twice under data analysis sections. Please check and delete where appropriate.

8. Please provide the Ethics approval number.

9. What is purpose of running a chi-square test and crude logistic regression as reported in table 2?

10. Table 2 is titled “Regression analysis results for ANC attendance” but reported Chi-square results in addition. I suggest the authors present the results of the logistics regression only in tables 2 and 3

11. How were confounding effects accounted for in establishing the association between the dependent and independent variables? Think the authors have to explore association in presence of other explanatory variables. This part of the analysis is not clear.

12. How was the Family type defined? how was the question asked?

13. In Figure 6, how is “Check-up” different from “Normal routine during pregnancy”? How were those two questions asked?

14. The statistical association cannot be strongly established with only crude results without accounting for explanatory variables. I suggest a revision of the statistical analysis to inform the conclusion of the study.

Reviewer #2: I thank the authors for the hard work and the efforts to contribute to a critical topic, especially in sub-Saharan Africa.

Kindly consider the following as they might improve the manuscript.

Topic:

The study seems to assess optimal utilization (four or more visits) of ANC among utilizers of ANC and not utilization of ANC as suggested by the topic.

Also, the authors should be adjusted the title to reflect that they only assessed socio-demographic factors.

Introduction:

Some of the text require references.

The justification is not compelling enough. According to you, ANC attendance is high but utilization for delivery is low. Isn't this a more interesting topic than exploring factors for ANC attendance? The authors need to better justify the investigation.

Conceptual framework:

Authors had a beautiful conceptual framework but limited the investigation to a few socio-demographic correlates of optimal ANC attendance. The reason for this is not clear.

The authors seem to suggest that attendance of four or more ANC visits is synonymous with utilization rather than optimal utilization.

Could the authors briefly explain the reasons for its (Anderson Behavioral Health care model) wide applicability? Why choose this model for the inquiry? Why not any other model?

Methods:

Study design:

I am not convinced that the method is adequate to answer the research question.

Assessing factors related to utilization of ANC services would be better answered by comparing utilizers and non-utilizers. Rather the design addresses correlates of optimum utilization (four or more) among utilizers. The authors need to clarify this.

Outcome variable:

How many outcome variables did this study have? The first sentence suggests more than one but itemizes just one. Kindly correct.

The authors should use “good” or “bad” knowledge for the sake of consistency and not knowledgeable or not.

The characterization of participants seems unclear. A part of the manuscript suggests that 50% was used as cut-off, while another part suggests that the participants’ mean score was used as cut-off. Kindly clarify and justify.

Validity and reliability of the study instrument:

How was construct validity assured?

What is the level of expertise of these researchers and clinicians? It may be helpful to state their qualification and give further background about their experience.

Questions 5, 6, 8, 9, 10 do not actually test what the women know?

Ethical consideration:

Could you include the reference number for the IRB approval?

Results:

Authors did not report results of bivariate analysis despite proposing to do them in the methods section. There is a need to report the salient findings even if all has been presented on table 2.

What is the mean age of the participants?

Authors need to define what is “far” or “long distance” concerning distance from residence to health facility.

The title of table 2 need to be updated to bivariate and regression analysis. The authors need to make some comments about results of bivariate analysis in prose.

Figure 3 is better presented as a pie chart.

On table 3 and age factor, if the reference group is the first group, then the Odds ration need to be reviewed.

For figure 6, kindly remove the bars with zero responses.

Discussion:

“The present finding of 69% is below the national coverage of at least four ANC visits of 75.9%”. Is there something about the study population that could account for this disparity?

“Our study finding is consistent with a study conducted in the southwest of Nigeria [39] but disagrees with a study conducted in Mozambique where the researchers found women’s knowledge of ANC services not to have significant importance in their utilisation of ANC services [40]. It is established that women with good knowledge of ANC services have a better understanding and acceptance of the services provided during ANC [41], and this knowledge will, therefore, be a catalyst for the utilising ANC services during pregnancy.” It seems more plausible that utilization of ANC services would result in better knowledge of ANC not the other way round as suggested by the authors. Anyway, the study is cross-sectional, hence the use of the word determinant is better replaced with association.

Limitations:

I can immediately think of some other limitations of this study e.g. cross-sectional nature, therefore, just associations can be measured, bias due to self-report, non-exploration of many other factors, etc. Kindly develop this section further.

Other comments”

I have made some further suggestions on the text of manuscript

Reviewer #3: PLOS ONE

Manuscript Title

Women’s knowledge of and factors associated with the utilization of antenatal care in

rural Ghana: analysis of a community-based cross-sectional study

Review

General Comments

Organization of the various sections need clarity and simplification of sentences. all sections must conform to the STROBE checklist for presenting prevalence studies..

Specific Comments

• Study Setting- is this a community based or facility based study? The title says community but the methods state facility based.

• Questionnaire administration- were the questionnaires self-administered or was administered by a research assistant or both?

• Table 2 needs to be simplified, Summarized; the detailed version can be attached as an appendix

• “Instead, the current study found women with low socioeconomic status more likely to

utilize ANC services at least four times due to their enrolment into the national health insurance

scheme” where was this result presented?

Are there any reasons why Bole had a lower than National ANC4+, What is the regional average?

6. PLOS authors have the option to publish the peer review history of their article (what does this mean?). If published, this will include your full peer review and any attached files.

Reviewer #1: No

Reviewer #2: Yes: Olumide ABIODUN

Reviewer #3: Yes: Dr. Alberta Amu

---

## [Author Response · Author response to Decision Letter 0]

1 May 2020

PONE-D-20-05929

Women’s knowledge and its associated factors regarding optimum utilisation of antenatal care in rural Ghana: a cross-sectional study 

Agani Afaya (MSc), Thomas B. Azongo (PhD), Veronica Millicent Dzomeku (PhD), Hyeonkyeong Lee (PhD), Richard Adongo Afaya (MPhil), Solomon Mohammed Salia (MSc), Peter Adatara (PhD), Robert Kaba Alhassan (PhD), 6Martin Amogre Ayanore (PhD), Abigail Kusi-Amponsah Diji, Confidence Alorse Atakro (MN), David Adadem (BN),Emmanuel Opoku Asiedu (BN), Paul Amuna (MB ChB, PhD)

Dear Editor,

We would like to thank you sincerely for the insightful reviewer comments on our manuscript and for the opportunity to resubmit the manuscript for another round of review. We find the comments very useful and have responded to them to the best of our knowledge. We acknowledge that the comments have no doubt helped improve the quality of our manuscript.

We herein provide further details by showing a point-by-point feedback on how each of the comments received were addressed. For easy identification, the reviewers’ comments have been repeated while Authors’ responses appear in BOLD text.

Amendment of Authorship

I would like to amend authorship to include Dr. Abigail Kusi-Amponsah Diji who contributed to conceiving and designing the study and contributed enormously to this paper. Her name has been included in the manuscript and highlighted in yellow ink. 

I apologise as this omission occurred during the initial submission process.

Thank you.

Review comments 

Reviewer #1

Review comments

1. Line numbering would have made it easy to referencing.

Authors response 

The Manuscript was formatted according to the Journal specifications. However, in view of the comment, line numbering has been included in the revised manuscript for easy referencing. 

Review comment

The statement “The sub-district has semi-urban area but predominantly rural” under study design section should be put under study settings.

Authors response 

Changes have been affected as suggested and now appears in the revised manuscript in page 8, lines 154-155.

Review comment

3. The variable “Number of Children” can’t be dichotomy when the categories are more than two. “The number of children was dichotomised into 1-2, 3-4 and 5+ children”

Authors response

The sentence has been revised: dichotomized has been changed to categorized. This is can be found in page 10, line 189 of the revised manuscript.

Review comments 

4. How was the distance to health facility measured? Distance from where to health facility? What distance is categorized as “far” and what is “near”?

Authors response 

For CHPS, the distance was between a radius of 5 km (GHS, 2002; MoH, 2016) and for Health Centres between 8 and 16 km radius (MEST, 2011). Women who trekked more than 5km and also 16km where considered far and those within the recommended limit were considered short. See page 10 lines 191-193.

[Ministry of Health, (2016). National Community-Based Health Planning and Services Policy. Ghana, Accra. 

Ghana Health Service (GHS), (2002). The Community-Based Health Planning and Services initiative: Concepts and plans for implementation. Accra, Ghana. 

Ministry of Environment Science and Technology (MEST), 2011. Zoning guidelines and planning standards. 

Review comments

5. “Those who sought ANC services less than four times were classified as having poor ANC service utilisation (inadequate) and those who had ANC visits of 4 or more were considered as having good (adequate) ANC service utilisation.” This information has been captured under the section on outcome variables already. Please delete it.

Authors response 

Repetition deleted under data analysis section see page 10, lines…

Review comment

6. Revise this statement: “Bivariate logistic regression analysis, computing odds ratio, was used to determine the strength of the association…” Because Logistic regression will report odds ratio. Bivariate logistic regression analysis reports more than strength of association. What about direction of the association?

Authors response

In page , lines this has been revised and now reads as; “Ordinary least square logistic regression analysis was applied where odds ratios were computed to determine how the independent variables is associated with the dependent variables (ANC service utilisation and knowledge of ANC)”

Review comments

7. The statement “ANC service utilisation was coded as 0 for poor utilisation (<4 ANC visits) and 1 for good untilisation (4+ ANC visits)” has been repeated under the outcome variables and twice under data analysis sections. Please check and delete where appropriate.

Authors response 

The repetition highlighted by the reviewer under data analysis have been deleted in the revised submission.

Review comments

8. Please provide the Ethics approval number.

Authors response 

This has been provided in page line and now reads as “Ethics approval for the study was obtained from the University of Health and Allied Sciences Research Ethical Committee (UHAS-REC A. 10[27] 17-18)”.

Review comments

9. What is purpose of running a chi-square test and crude logistic regression as reported in table 2?

Authors response 

Chi-square results has been removed and replaced with tables only highlighting analysis of logistic regression (for adjusted and unadjusted)

Review comments

10. Table 2 is titled “Regression analysis results for ANC attendance” but reported Chi-square results in addition. I suggest the authors present the results of the logistics regression only in tables 2 and 3

Authors response

As suggested by the reviewer, authors have re-presented the results of the logistics regression only in tables 2 and 3.

Review comments

11. How were confounding effects accounted for in establishing the association between the dependent and independent variables? Think the authors have to explore association in presence of other explanatory variables. This part of the analysis is not clear.

Authors response 

National health insurance was controlled for in the adjusted model. When NHIS was controlled for in the adjusted model there was a significant relationship with optimum ANC and these explanatory variables (Age, marital status, educational level, religion, and distance to health facility). There for NHIS was found to have confounder effect when included in the regression model. 

Review comments

12. How was the Family type defined? how was the question asked?

Authors response

Nuclear family was termed as a family group consisting of two parents and their children (one or more). Extended family: consisting of parents like father, mother, and their children, aunts, uncles, grandparents, and cousins, all living in the same household. 

Do you live with your husband and children alone or with your other extended family (grandfather etc)

Review comments

13. In Figure 6, how is “Check-up” different from “Normal routine during pregnancy”? How were those two questions asked?

Authors response 

Check-up: the main reason behind they attending ANC was to check their health status and that of their unborn baby. “Normal routine during pregnancy” mothers that were categorized here where those who mainly said their reason for attending ANC was because it is the routine thing to do during pregnancy and has been asked to come on this date by nurses.

Review comments

14. The statistical association cannot be strongly established with only crude results without accounting for explanatory variables. I suggest a revision of the statistical analysis to inform the conclusion of the study.

Authors response 

Reviewer concerns have been addressed in table 2 and 3

Reviewer # 2

Review comments

The study seems to assess optimal utilization (four or more visits) of ANC among utilizers of ANC and not utilization of ANC as suggested by the topic. 

Authors response 

Taking the reviewer suggestion into consideration, authors have revised the title as found in the revised manuscript submitted. See page 1 lines 2-3

Review comments

Also, the authors should be adjusted the title to reflect that they only assessed socio-demographic factors.

Authors response 

While acknowledging that this is a fair comment from reviewers, other factors outside individual socio-demographic factors were assessed i.e community and individual challenges affecting antenatal utilization; facility related barriers to ANC attendance; reasons for ANC Attendance etc. Hence, adjusting the title to reflect socio-demographic factors has a potential to limit the scope of factors considered in this study. 

Given the title change in this revised submission, authors think we have adequately addressed any concerns regarding the suitability of the title and that the title now reflects the aim/objectives of the study.

Introduction:

review comment 

Some of the text require references.

Authors response

References have been provided in relevant areas in the introduction section. 

Review comment

The justification is not compelling enough. According to you, ANC attendance is high but utilization for delivery is low. Isn't this a more interesting topic than exploring factors for ANC attendance? The authors need to better justify the investigation.

Authors response 

This has been addressed in page 4 and 5 lines 89-104. The highlighted portions provide further clarity on areas these have been addressed. 

Review comments

Conceptual framework:

Authors had a beautiful conceptual framework but limited the investigation to a few socio-demographic correlates of optimal ANC attendance. The reason for this is not clear.

The authors seem to suggest that attendance of four or more ANC visits is synonymous with utilization rather than optimal utilization.

Authors response

We adapted the behavioural model framework of Andersen for use of health services to identify the factors that potentially facilitate or impede minimum number of antenatal health services seeking behavior at individuals and community levels.

In considering the review suggestion, authors have made changes and have maintained optimal utilisation instead of utilisation in this revised submission of the manuscript. 

Review comments

Could the authors briefly explain the reasons for its (Anderson Behavioral Health care model) wide applicability? Why choose this model for the inquiry? Why not any other model?

Authors response

The frame work is widely used because the three factors identified by Andersons model to facilitate or impede healthcare access cuts across almost every health system across the globe including the developed and developing countries. And several studies have used the model to identify facilitating and impeding factors to healthcare access. 

The framework predicts that a series of factors predisposing, enabling and need factors influence the utilization of health services by people. According to the model, predisposing factors are demographics and social structures. Enabling factors facilitates individuals to use services for example, availability of resources such as income, access to free services, availability and access to the service. Need factors motivates service use. We chose this model because we sought to determine factors that impede or facilitate ANC services utilization in health facilities. And Anderson behavioral model best suit our current study objective. 

Methods:

Review comments

Study design:

I am not convinced that the method is adequate to answer the research question.

Assessing factors related to utilization of ANC services would be better answered by comparing utilizers and non-utilizers. Rather the design addresses correlates of optimum utilization (four or more) among utilizers. The authors need to clarify this.

Authors response 

Reviewer suggestion and comments was helpful in making revisions to the design section of the revised manuscript. This study truly addresses correlates of optimum utilization among users and this has reflected in the revised manuscript under the study design section. See page 8 lines 137-140.

Outcome variable:

Review comments

How many outcome variables did this study have? The first sentence suggests more than one but itemizes just one. Kindly correct.

Authors response 

The outcome variables were optimum utilisation and Knowledge of ANC. This has been addressed under outcome variables. See page 9 lines 172-173.

Review comments

The authors should use “good” or “bad” knowledge for the sake of consistency and not knowledgeable or not.

Authors response 

Authors acknowledge this as a fair suggestion. However, to ensure consistency in the revised manuscript, authors have used good and poor knowledge. 

Review comments

The characterization of participants seems unclear. A part of the manuscript suggests that 50% was used as cut-off, while another part suggests that the participants’ mean score was used as cut-off. Kindly clarify and justify.

Authors response

Authors acknowledge the inconsistent reporting in the methods section with regard to the cut-off point in knowledge assessment. Author has corrected this inconsistency by deleting the 50% cut-off point.

Review comments

Validity and reliability of the study instrument:

How was construct validity assured?

Authors response

construct validity pertains to a specific use of a scale and can often be context or population dependent. The current study instrument was contextually structured to identify factors that facilitate or impede optimal utilization of ANC services within the study population. 

Multi-collinearity diagnostics were done and found that the Variance Inflation Factors (VIFs) were all below the 5-10 rule of thumb range, suggesting there is no collinearity among the independent variables to be fitted in the regression model! Effects of multi-collinearity was therefore ruled out and did not have any effect on the validity of the study outcomes.

Review comments

What is the level of expertise of these researchers and clinicians? It may be helpful to state their qualification and give further background about their experience.

Authors response

Thomas B. Azongo (PhD) is a snr lecturer and a public health expert; Veronica Millicent Dzomeku (PhD) is a midwife and a Snr Lecturer, Hyeonkyeong Lee (PhD) is a professor in community health nursing; Peter Adatara (PhD) is a snr lecturer and his specialty area is in maternal and child health; Robert Kaba Alhassan (PhD), Snr research fellow; Martin Amogre Ayanore (PhD) is a Health Economist and a Public health expert, Paul Amuna (MB ChB, PhD) is a professor and his area of specialty is maternal and child health 

Clinicians: They both have degree in midwifery and have been practicing midwifery for over 10 years 

Review comments 

Questions 5, 6, 8, 9, 10 do not actually test what the women know?

Authors response

These questions were added to assess mothers knowledge on the other activities carried during ANC: Mothers that utilize ANC are taught these and they are expected to have basic knowledge in relation to the education or things taught during ANC attendance: for example during ANC, mothers are informed of the recommended ANC visits, so we expected mothers to know these. Additionally, mothers who utilized ANC are educated on the recommended place for safe delivery and the necessary items required for delivery. The items for assessing the mother’s knowledge of ANC was developed in context with the study population

Review comments

Ethical consideration:

Could you include the reference number for the IRB approval?

Authors response

Ethics approval for the study was obtained from the University of Health and Allied Sciences Research Ethical Committee (UHAS-REC A. 10[27] 17-18).

Results:

Review comments

Authors did not report results of bivariate analysis despite proposing to do them in the methods section.

Authors response

The results section has been updated and review concerns above addressed. See table 2 and 3 on pages 16 &17 lines 284-285

Review comment

There is a need to report the salient findings even if all has been presented on table 2.

Authors response

Authors acknowledge this and have done the needful by focusing on only the salient findings. See page 15 lines 278-283.

Review comment 

What is the mean age of the participants?

Authors response 

Age of participants were categorized during data collection therefore the mean age could not be calculated.

Review comments

The title of table 2 need to be updated to bivariate and regression analysis. The authors need to make some comments about results of bivariate analysis in prose.

Authors response

Table 2 and 3 have been updated. 

Review comments

Figure 3 is better presented as a pie chart.

Authors response

Figure 3 has been changed to a pie chart. Refer to page 18.

Review comment

On table 3 and age factor, if the reference group is the first group, then the Odds ratio need to be reviewed.

Authors response 

This critical suggestion was reviewed and significant changes made to the odds ratios. Table 3 provide further details on the revised submission after addressing this concerns. See page 20.

Review comments

For figure 6, kindly remove the bars with zero responses.

Authors response

Changes has been made as suggested. Refer to page 14

Discussion:

Review comment

“The present finding of 69% is below the national coverage of at least four ANC visits of 75.9%”. Is there something about the study population that could account for this disparity?

Authors response 

 This has been addressed in page 25 line 348-351 and now reads as;

This difference may be due to the lack of awareness of optimal ANC visits. Also, the differences could have been caused by other potential inhibiting factors such as distance to health facilities and socio-economic factors. 

Review comments 

“Our study finding is consistent with a study conducted in the southwest of Nigeria [39] but disagrees with a study conducted in Mozambique where the researchers found women’s knowledge of ANC services not to have significant importance in their utilisation of ANC services [40]. It is established that women with good knowledge of ANC services have a better understanding and acceptance of the services provided during ANC [41], and this knowledge will, therefore, be a catalyst for the utilising ANC services during pregnancy.” It seems more plausible that utilization of ANC services would result in better knowledge of ANC not the other way round as suggested by the authors. Anyway, the study is cross-sectional, hence the use of the word determinant is better replaced with association.

Authors response

We totally agree with assertion (It seems more plausible that utilisation of ANC services would result in better knowledge of ANC). In the discussion section, this emphasizes has been made in relation to the findings of the study.

Limitations:

Review comments

I can immediately think of some other limitations of this study e.g. cross-sectional nature, therefore, just associations can be measured, bias due to self-report, non-exploration of many other factors, etc. Kindly develop this section further.

Authors response

A section on limitations has been added and some potential limitation of the study acknowledged. See page 29 lines 448-458

Review comments 

Other comments”

I have made some further suggestions on the text of manuscript

Authors response

Review concerns on inconsistencies in the manuscript have been addressed. Authors read thoroughly through the revised manuscript to identify these inconsistencies and sentences that were not clear and corrections effected. 

Reviewer #3: 

Manuscript Title

Women’s knowledge of and factors associated with the utilization of antenatal care in

rural Ghana: analysis of a community-based cross-sectional study

Review

General Comments

Organization of the various sections need clarity and simplification of sentences. all sections must conform to the STROBE checklist for presenting prevalence studies.

Authors response

We acknowledge and used the STROBE checklist to guide the study (methods)

Specific Comments

Review comments

• Study Setting- is this a community based or facility-based study? The title says community but the methods state facility based.

Authors response

The title has been revised to reflect these concerns. See page 1

Review comment

• Questionnaire administration- were the questionnaires self-administered or was administered by a research assistant or both?

Authors response

The questionnaires were administered by only the four research assistants trained for data collection 

Review comment

• Table 2 needs to be simplified, Summarized; the detailed version can be attached as an appendix.

Authors response

Authors acknowledge and have addressed it. Authors reported on only salient findings of the study but not every variable see table 2 on pages 16 & 17 lines 284-285.

Review comment

• “Instead, the current study found women with low socioeconomic status more likely to

utilize ANC services at least four times due to their enrolment into the national health insurance

scheme” where was this result presented?

Authors response

This was a general assumption for the study participants. This have been deleted because it lacked evidence from the current study.

Review comments 

Are there any reasons why Bole had a lower than National ANC4+, What is the regional average?

Authors response 

The difference may be due to the lack of awareness of optimal ANC visits. Also, the differences could have been caused by other potential inhibiting factors such as distance to health facilities and socio-economic factors. Refer to page 25 line 348-351.

---

## [Decision Letter · Decision Letter 1]

25 May 2020

PONE-D-20-05929R1

Women’s knowledge and its associated factors regarding optimum utilisation of antenatal care in rural Ghana: a cross-sectional study

PLOS ONE

Dear Agani Afaya,

Thank you for submitting your manuscript to PLOS ONE. After careful consideration, we feel that it has merit but does not fully meet PLOS ONE’s publication criteria as it currently stands. Therefore, we invite you to submit a revised version of the manuscript that addresses the points raised during the review process.

We look forward to receiving your revised manuscript.

Kind regards,

Prof Kwasi Torpey, MD PhD MPH

Academic Editor

PLOS ONE

Reviewers' comments:

Reviewer's Responses to Questions

**Comments to the Author**

1. If the authors have adequately addressed your comments raised in a previous round of review and you feel that this manuscript is now acceptable for publication, you may indicate that here to bypass the “Comments to the Author” section, enter your conflict of interest statement in the “Confidential to Editor” section, and submit your "Accept" recommendation.

Reviewer #3: (No Response)

2. Is the manuscript technically sound, and do the data support the conclusions?

Reviewer #3: Yes

3. Has the statistical analysis been performed appropriately and rigorously? 

Reviewer #3: Yes

4. Have the authors made all data underlying the findings in their manuscript fully available?

Reviewer #3: Yes

5. Is the manuscript presented in an intelligible fashion and written in standard English?

Reviewer #3: Yes

6. Review Comments to the Author

Reviewer #3: Line 146 do you mean a male practicing midwife? Correct

Line 193- 194 disjointed sentence, Reconsider

Line 198 4 trained RAs assisted or were assisted? Clarify

Line 196 still has ‘A self-administered questionnaire’

Line 272 correct tha

Word Usage

Line 413 incomplete---optimum? needed?

Line 424 effect or importance? reconsider

7. PLOS authors have the option to publish the peer review history of their article (what does this mean?). If published, this will include your full peer review and any attached files.

Reviewer #3: No

---

## [Author Response · Author response to Decision Letter 1]

26 May 2020

PONE-D-20-05929

Women’s knowledge and its associated factors regarding optimum utilisation of antenatal care in rural Ghana: a cross-sectional study 

Agani Afaya (MSc), Thomas B. Azongo (PhD), Veronica Millicent Dzomeku (PhD), Hyeonkyeong Lee (PhD), Richard Adongo Afaya (MPhil), Solomon Mohammed Salia (MSc), Peter Adatara (PhD), Robert Kaba Alhassan (PhD), 3Abigail Kusi-Amponsah Diji, Confidence Alorse Atakro (MN), David Adadem (BN),Emmanuel Opoku Asiedu (BN), Paul Amuna (MB ChB, PhD), Martin Amogre Ayanore (PhD),

Dear Editor,

We would like to thank you sincerely for the reviewer comments on our manuscript and for the opportunity to resubmit the manuscript for publication consideration. We find the comments very useful and have responded to them. We acknowledge that the comments have no doubt helped improve the quality of our manuscript.

We herein provide further details by showing a point-by-point feedback on how each of the comments received were addressed. For easy identification, the reviewers’ comments have been repeated while Authors’ responses appear in yellow text.

Review comments 

Reviewer #3: Line 146 do you mean a male practicing midwife? Correct

Authors response

Yes, a male practicing midwife: changes have been effected see line 146

Reviewer comments

Line 193- 194 disjointed sentence, Reconsider

Authors response

The sentence has been restructured and it reads as: Other variables of interest were distance to health facility (For CHPS, the distance was between a radius of 5 km (and for Health Centres between 8 and 16 km radius [20-22]. Women who trekked more than 5km and also 16km where considered far and those within the recommended limit were considered short), and means of transportation to the facility. 

Reviewer comment

Line 198 4 trained RAs assisted or were assisted? Clarify

Authors response

Four (4) trained RAs assisted to collect data in the four study health facilities: see line 198

Reviewer comment

Line 196 still has ‘A self-administered questionnaire’

Authors response

Correction made and self-administered deleted. See line 196

Reviewer comments

Line 272 correct tha

Authors response

This has been corrected, thank you

Word Usage

Reviewer comments

Line 413 incomplete---optimum? needed?

Authors response

Correction made as requested

Reviewer comments

Line 424 effect or importance? Reconsider

Authors response

Author agree with your suggestion; importance has been changed to effect

---

## [Editor Report · Decision Letter 2]

29 May 2020

Women’s knowledge and its associated factors regarding optimum utilisation of antenatal care in rural Ghana: a cross-sectional study

PONE-D-20-05929R2

Dear Agani Afaya,

We are pleased to inform you that your manuscript has been judged scientifically suitable for publication and will be formally accepted for publication once it complies with all outstanding technical requirements.

With kind regards,

Professor Kwasi Torpey, MD PhD MPH

Academic Editor

PLOS ONE
---

## [Editor Report · Acceptance letter]

23 Jun 2020

PONE-D-20-05929R2 

Women’s knowledge and its associated factors regarding optimum utilisation of antenatal care in rural Ghana: a cross-sectional study 

Dear Dr. Afaya:

I'm pleased to inform you that your manuscript has been deemed suitable for publication in PLOS ONE. Congratulations! Your manuscript is now with our production department. 

Kind regards, 

on behalf of

Professor Kwasi Torpey 

Academic Editor

PLOS ONE